# Dissipative Landau-Zener tunneling in the crossover regime from weak to strong environment coupling

X. Dai [1,2,16] ✉, R. Trappen [1,2,16] ✉, H. Chen [3,4], D. Melanson[1,2], M. A. Yurtalan[1,2,5], D. M. Tennant[1,2], A. J. Martinez [1,2], Y. Tang [1,2], E. Mozgunov[6], J. Gibson[7,8], J. A. Grover [7,9], S. M. Disseler[7,10], J. I. Basham[7,11], S. Novikov [7,12], R. Das[10], A. J. Melville[10], B. M. Niedzielski [10], C. F. Hirjibehedin [10], K. Serniak[10], S. J. Weber[10], J. L. Yoder[10], W. D. Oliver[9,10], K. M. Zick[6,7], D. A. Lidar [3,4,13,14] & A. Lupascu [1,2,15] ✉

Landau-Zener tunneling, which describes the transition in a two-level system during a sweep through an anti-crossing, is a model applicable to a wide range of physical phenomena. Realistic quantum systems are affected by dissipation due to coupling to their environments. An important aspect of understanding such open quantum systems is the relative energy scales of the system itself and the system-environment coupling, which distinguishes the weak- and strong-coupling regimes. Using a tunable superconducting flux qubit, we observe the crossover from weak to strong coupling to the environment in Landau-Zener tunneling. Our results confirm previous theoretical studies of dissipative Landau-Zener tunneling in the weak and strong coupling limits. We devise a spin bath model that effectively captures the crossover regime. This work is relevant for understanding the role of dissipation in quantum annealing, where the system is expected to go through a cascade of Landau-Zener transitions before reaching the target state.

Landau-Zener (LZ) tunneling[1–4] describes non-adiabatic transitions through an anti-crossing in a two-state quantum system with a linearly changing energy separation. The LZ model is applicable to a wide range of physical phenomena, such as atomic collisions[5], chemical reactions[6], and molecular magnets[7]. Physical realizations of LZ tunneling are influenced by system-environment coupling[8–12], and there are extensive theoretical studies on the effects of dissipation on LZ tunneling[13–21].

Studying the effects of dissipation in quantum systems is of both fundamental interest[22,23] and important for practical applications such as quantum information processing[24]. Dissipation is of particular concern to analog quantum computation, where error due to dissipation cannot be indefinitely suppressed due to the lack of a fault-tolerance threshold[24]. One prominent analog quantum algorithm is quantum annealing[25–27], where a system goes through one or multiple

[1]Institute for Quantum Computing, University of Waterloo, Waterloo, ON, Canada. [2]Department of Physics and Astronomy, University of Waterloo, Waterloo, ON, Canada. [3]Center for Quantum Information Science & Technology, University of Southern California, Los Angeles, CA, USA. [4]Department of Electrical & Computer Engineering, University of Southern California, Los Angeles, CA, USA. [5]Department of Electrical and Computer Engineering, University of Waterloo, Waterloo, ON, Canada. [6]University of Southern California—Information Sciences Institute, Arlington, VA, USA. [7]Northrop Grumman Corporation, Linthicum, MD, USA. [8]Department of Physics and Astronomy, Dartmouth College, Hanover, NH, USA. [9]Research Laboratory of Electronics, Massachusetts Institute of Technology, Cambridge, MA, USA. [10]Lincoln Laboratory, Massachusetts Institute of Technology, Lexington, USA. [11]QuEra Computing Inc., Boston, MA, USA. [12]Atlantic Quantum Corp., Cambridge, MA, USA. [13]Department of Chemistry, University of Southern California, Los Angeles, CA, USA. [14]Department of Physics, University of Southern California, Los Angeles, CA, USA. [15]Waterloo Institute for Nanotechnology, University of Waterloo, Waterloo, ON, Canada. [16]These authors contributed equally: X. Dai, R. Trappen. ✉e-mail: x35dai@uwaterloo.ca; rtrappen@uwaterloo.ca; adrian.lupascu@uwaterloo.ca

LZ tunnelings to reach some many-body quantum state of interest, relevant to quantum simulation or optimization tasks. The role of dissipation in quantum annealing is still largely an open question, with previous studies suggesting that dissipation could either improve the annealing performance by cooling the system[28–30] or hamper the tunneling process[31–33], depending on the relative strength between the system-environment coupling and relevant system energy scales.

For the two-state LZ tunneling with a single sweep across the anti-crossing, analytical solutions for the transition probabilities in the closed-system case were obtained in refs. [1–4]. Specifically, for a given tunneling amplitude (or equivalently the minimum spectral gap size) $\Delta$ and sweep velocity $v$, the probability for the system to tunnel to the different diabatic state is $1 - P_{LZ}$, where $P_{LZ}$ in the coherent limit is given by

$$P_{LZ} = \exp\left(-\frac{\pi\Delta^2}{2\hbar v}\right). \tag{1}$$

For most devices made for quantum information processing tasks, dissipation is well described by assuming weak coupling to a Markovian environment. Applying these assumptions to the Landau-Zener problem and considering noise coupled longitudinally (or equivalently to the diabatic states), it has been found that the dependence of the transition probability on sweep rate is unchanged if the noise temperature is low compared to the system's minimum spectral gap at the anti-crossing and approaches the asymptotic value of 1/2 when the noise temperature is high[14]. At intermediate temperatures, it has been predicted that the noise leads to non-monotonic dependence of the tunneling probability on the sweep rate, due to the competition between adiabaticity, thermal excitation near the anti-crossing, and thermal relaxation after the anti-crossing[19]. This non-monotonic dependence has not yet been observed, due to the intricate requirement on the various energy scales.

Strong system-environment coupling occurs both naturally in physical phenomena such as electron transfer[34] and in engineered quantum devices such as quantum annealers[35]. A general description in the strong coupling regime is challenging due to finite system-bath correlations and non-Markovian effects. Here we focus on strong low-frequency noise, which is ubiquitous in solid-state systems such as superconducting qubits[36,37]. Superconducting flux qubits are one of the leading platforms for quantum annealing, and the dominant noise is the intrinsic flux noise[38], the physical origin of which remains elusive to date. While the noise power spectral density of flux noise has been well-characterized[39–41], a general open-system model for flux noise remains challenging due to its long correlation time. In the specific case of quantum tunneling with small tunneling amplitude, it has been shown in macroscopic resonant tunneling (MRT) experiments that longitudinally coupled low-frequency noise can be modeled as an environmental polarization that preferentially aligns with the diabatic states of the system[42,43], which can be formally described using the polaron transformation[32,44]. For Landau-Zener tunneling, the transition probability in the presence of strong low-frequency noise remains unchanged from the coherent case, provided that the integrated noise amplitude is higher than the noise temperature[45].

In this work, we experimentally investigate LZ tunneling with a tunable flux qubit for a wide range of minimum gap $\Delta$ and sweep velocity $v$. We present detailed numerical modeling of the open-system effects, which goes beyond the phenomenological models adopted in previous experiments[8,10,46]. Comparing the experimental and numerical models reveals that as the tunneling amplitude $\Delta$ decreases, the dominant open-system effects display a crossover from that of a weakly-coupled Markovian environment to a strongly coupled non-Markovian environment. We further devise a toy environment model in terms of a spin bath, which shows qualitative agreement with the experiment for the full range of parameters. The spin bath model

suggests that the crossover arises as the time scale for the qubit to complete the tunneling, determined by $\Delta$, becomes comparable to the time scale over which the low-frequency environment reorganizes itself into the lower energy configuration. Our experiment probes a previously unexplored regime of system-environment coupling and the theoretical analysis presents new directions toward understanding and modeling intrinsic flux noise in superconducting circuits, which is both of fundamental importance and practical value for assessing the capability of analog quantum processors such as the quantum annealer.

## Results

### The tunable capacitively-shunted flux qubit

Our experiments are performed using a two-level quantum system implemented using a tunable superconducting capacitively-shunted flux qubit[47–49]. A schematic of the experiment setup is shown in Fig. 1a. The qubit circuit consists of two flux loops, designated as $z$, $x$ respectively, including Josephson tunnel junctions. Under suitable flux bias conditions, the circuit has a double-well potential energy landscape, with the two wells corresponding to persistent current flowing in opposite directions in the $z$ loop. When cooled down to the base temperature of a dilution fridge, the system is confined to the respective ground states of the two potential wells, described by the two-state (qubit) Hamiltonian

$$H_q = -\frac{\epsilon(\Phi_z)}{2}\sigma_z - \frac{\Delta(\Phi_x)}{2}\sigma_x, \tag{2}$$

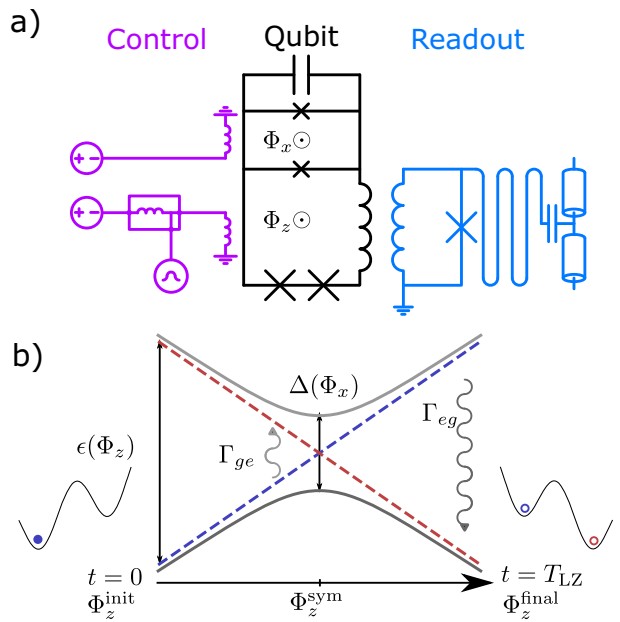

**Fig. 1 | The flux qubit and the dissipative LZ transition. a** Schematic of the tunable capacitively-shunted flux qubit and the control and readout circuitry. The flux biases $\Phi_x$, $\Phi_z$ are each supplied by a DC voltage source and the $\Phi_z$ is further controlled by a fast arbitrary waveform generator, joined to the DC control through a diplexer. Readout is done by measuring the transmission through an rf-SQUID terminated waveguide resonator coupled inductively to the qubit. **b** Schematic representation of the LZ sequence. The blue and red dashed lines indicate the energies of the diabatic states, which are separated by $\epsilon(\Phi_z) = 2I_p(\Phi_z - \Phi_z^{sym})$. The grey lines indicate the eigenenergies of the qubit, which has a minimum gap of $\Delta$ at the symmetry point $\Phi_z^{sym}$. The curly arrows indicate the dominant open-system effects in the LZ measurements in the weak-coupling limit, which are excitations around the symmetry point and relaxation after the symmetry point. The double-well plots on either side of the energy level diagram are a representation of the qubit potential at the beginning and end of the LZ sweep.

with $\sigma_{z,x}$ being the Pauli operators. Here $\epsilon = 2I_p(\Phi_z - \Phi_z^{sym})$ and $\Delta$ are respectively the bias and tunneling amplitude between persistent current states, $I_p$ is the persistent current, $\Phi_{z(x)}$ is the flux bias in the $z(x)$ loop, and $\Phi_z^{sym}$ is the $\Phi_z$ bias which gives a symmetric double-well potential. The $\Phi_x$, $\Phi_z$ biases are controlled by DC voltage sources and the $\Phi_z$ bias is additionally coupled to a fast arbitrary waveform generator (AWG), combined with the DC control through a diplexer. Readout of persistent current states is done by inductively coupling the qubit $z$ loop to a flux-sensitive resonator. The circuit is capacitively coupled to a waveguide used to send microwave signals, allowing resonant excitation of the circuit. A circuit network model is fit to the spectroscopically measured transition frequencies, which allows for determining the circuit parameters. The two-state model parameters $I_p$ and $\Delta$ can then be obtained from the circuit model at arbitrary flux biases near the symmetry point $\Phi_z = \Phi_z^{sym}$. Details of the experimental setup and calibration measurements are given in Supplementary Notes 1–6.

## LZ tunneling in the short-time limit

A diagrammatic representation for the LZ measurement is shown in Fig. 1b. At $t = 0$, the qubit is prepared in the left well at $\Phi_z^{init} \approx -0.005\Phi_0 + \Phi_z^{sym}$. Then the qubit $z$ flux $\Phi_z$ is linearly swept to $\Phi_z^{final} \approx 0.005\Phi_0 + \Phi_z^{sym}$ over the duration $T_{LZ}$. The initial and final values of $\Phi_z$ ensure the LZ sweep starts and ends far enough from the anti-crossing so that the qubit energy eigenstates approximately overlap with the persistent current states. The qubit state population after the sweep is read out by measuring the state-dependent transmission through the resonator. The sequence is repeated for a range of $\Phi_x$ and $T_{LZ}$ values. With decreasing $\Phi_x$, $\Delta$ decreases nearly exponentially whereas $I_p$ increases by about 10% over the entire range.

The measured final excited state probabilities $P_e$ versus $T_{LZ}$ at short times are shown in Fig. 2a. In the weak-coupling limit, the system is expected to behave nearly coherently for short sweep times, implying that the final excited state probabilities are well described by Eq. (1), with the sweep velocity given by

$$v = \frac{2I_p(\Phi_z^{final} - \Phi_z^{init})}{T_{LZ}}. \tag{3}$$

To confirm the coherent-limit behavior, we fit an exponential decay to the short-time decay of the measured final excited state probabilities and then convert the decay constant to an effective gap $\Delta_{LZ}$ assuming $I_p$ given by the circuit model. The extracted values of the effective gap $\Delta_{LZ}$ are compared to values of $\Delta$ given by the circuit model in Fig. 2b. There is excellent agreement for the full range of $\Phi_x$ measured in LZ experiments, with $\Delta$ in the range of 12–120 MHz. Further increasing or decreasing $\Delta$ would prevent us from observing the full transition from non-adiabatic to adiabatic evolution within the range of sweep time afforded by our flux control bandwidth.

## LZ tunneling in the long-time limit

After confirming the short-time behavior, we observe the dynamics at longer sweep times, where coupling to the environment is expected to affect the LZ transition. We first discuss the characterization of the environment. Measurement of the noise spectrum is done based on its effect on qubit relaxation and dephasing at $\Delta/h \gtrsim 1\,\text{GHz}$[50], where the weak coupling limit holds. We find that the coherence is flux noise limited and can be explained by a noise power spectral density (PSD) consistent with previous work[40,47,51], where the noise power varies to a good approximation as $1/f$, with $f$ the frequency, up to 1 GHz, combined with quasi-ohmic noise at higher frequencies. Our noise measurements, which are sensitive to the symmetrized noise power, combined with the assumption that the environment is in thermal equilibrium at the fridge base temperature, allow us to write the quantum noise PSD

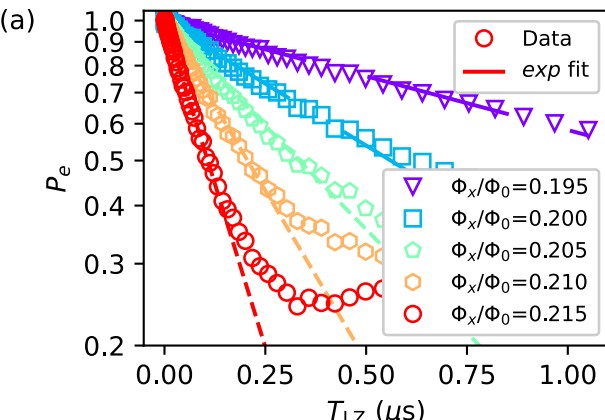

(a)

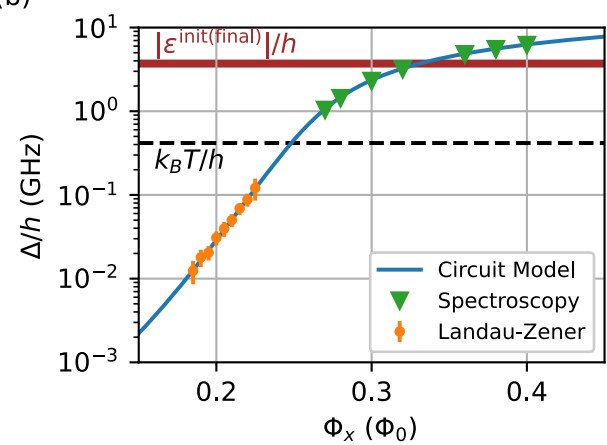

(b)

**Fig. 2 | LZ data in the short sweep time range and fitted effective gap.**
**a** Experimental data (open markers) and exponential fits (dashed lines) to the final excited states $P_e$ versus the sweep time $T_{LZ}$. The maximum sweep time included in the fit is determined adaptively, by first starting from 30 ns and then increasing until the mean square loss of the fit exceeds 0.01. **b** The minimum gap $\Delta$ versus x-bias flux $\Phi_x$ from spectroscopy (green triangle), LZ (orange dots), and circuit model (blue line). Error bars in the LZ data represent standard error propagated from the exponential decay fit error. The circuit model is a result of fitting spectroscopy data for a range of $\Phi_x$, $\Phi_z$ (not shown here). The black dashed line indicates the noise temperature, which is assumed to be close to the base temperature of the dilution fridge, $T = 20$ mK. The red horizontal band indicates the qubit energy splitting at the beginning and the end of the LZ sweep.

(unsymmetrized) as

$$S_\lambda(\omega) = S_{\lambda,1/f}(\omega) + S_{\lambda,\text{ohmic}}(\omega), \tag{4}$$

$$S_{\lambda,1/f}(\omega) = \frac{A_\lambda \omega}{|\omega|^\alpha}\left[1 + \coth\left(\frac{\beta\hbar\omega}{2}\right)\right] \text{ and} \tag{5}$$

$$S_{\lambda,\text{ohmic}}(\omega) = B_\lambda \omega|\omega|^{\gamma-1}\left[1 + \coth\left(\frac{\beta\hbar\omega}{2}\right)\right], \tag{6}$$

with $\lambda \in \{\Phi_x, \Phi_z\}$. Here $\beta = 1/k_BT$ is the inverse temperature and $A_\lambda(B_\lambda)$ and $\alpha(\gamma)$ characterize the amplitude and frequency dependence of the $1/f$(quasi-ohmic) component respectively. Given the smaller noise power and coupling matrix elements of the $\Phi_x$ noise for flux biases probed in the LZ measurement, we only consider $\Phi_z$ noise from here onward (see Supplementary Note 7).

The measured final ground state probabilities $P_g = 1 - P_e$ for different $\Phi_x$ are shown in Fig. 3a, b, versus the full range of sweep time $T_{LZ}$ and the dimensionless time $\tau = \Delta^2/\hbar v$, with $\Delta$ being the predicted value

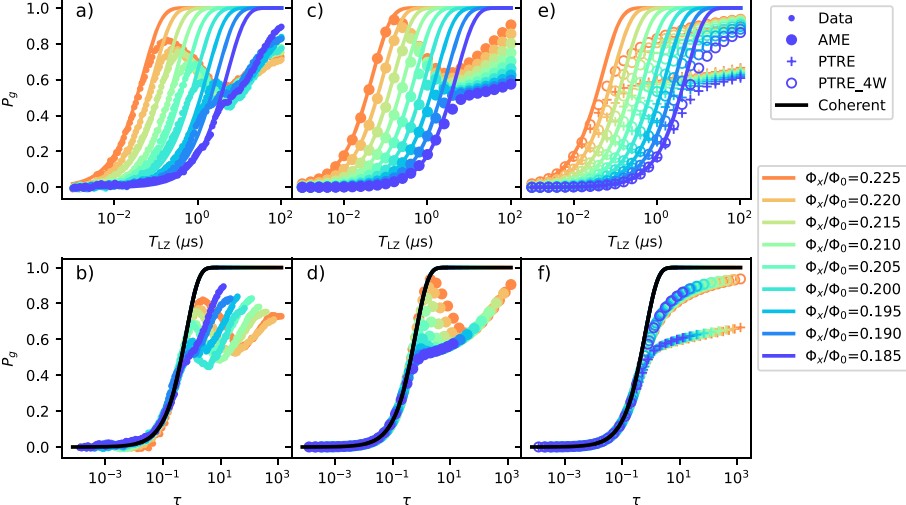

**Fig. 3 | LZ data for the full range of $\Phi_x$ and sweep time, and comparison with simulation.** Final ground-state probability $P_g$ versus the sweep time $T_{LZ}$ (top) and the dimensionless sweep time $\tau = \Delta^2/\hbar\nu$ (bottom), for (**a, b**) experimental data, **c, d** Adiabatic master equation simulation (labeled as AME) results with nominal noise parameters and (**e,f**) Polaron-transformed Redfield equation simulation with nominal noise parameters (labeled as PTRE) and 4 times larger MRT width $W$ (labeled as PTRE_4W). All panels also contain the coherent limit given by $P_g = 1 - P_{LZ}$.

from the circuit model, and $\nu$ the sweep velocity defined in Eq. (3). Analyzing the dependence on both the actual time $T_{LZ}$, and dimensionless time $\tau$ allows us to make complementary observations about the changes in the effect of the environment as $\Phi_x$, or equivalently $\Delta$, is tuned.

## Simulation with master equations in the weak- and strong-coupling limit

The weak coupling limit between the system and the environment is expected to apply when the system-environment coupling strength is much smaller than the system's own energy scale, in our case $\Delta$. In the weak coupling limit, the adiabatic master equation (AME)[52] can be applied, where the environmental effect is assumed to be Markovian and leads to thermal transitions and decoherence between the energy eigenstates of the system. The AME-simulated final ground state probabilities are shown in Fig. 3c, d. For large $\Phi_x$, $P_g$ increases non-monotonically with $T_{LZ}$. This can be interpreted in terms of the competition between thermal excitation around the minimum gap and relaxation after the minimum gap at intermediate and long-time scales, as seen in previous numerical studies of dissipative LZ[19,20,53]. As the total sweep time increases, $P_g$ first increases following the coherent limit, until the qubit has enough time to thermalize around the minimum gap and $P_g$ starts to decrease toward 0.5, as $k_BT \gg \Delta$. Further increase of the total sweep time allows the qubit to relax after crossing the minimum gap, which leads to increasing $P_g$, as $k_BT \ll \epsilon^{\text{final}} \equiv 2I_p(\Phi_z^{\text{final}} - \Phi_z^{\text{final}})$. Since the instantaneous matrix element of the $\Phi_z$ noise is proportional to $\Delta/\sqrt{\Delta^2 + \epsilon(t)^2}$, thermal relaxation after the minimum gap is slow, as $\epsilon^{\text{final}} \gg \Delta$, and is only significant at very long sweep times.

Comparing the experimental data and AME simulation, we find qualitative agreement at large $\Phi_x$. The experimental data show smaller $P_g$ local maxima, which could indicate that the noise experienced by the qubit in the LZ sweep is larger than the extrapolated noise values based on qubit decoherence measurements. We note that at $\Phi_x = 0.2$, the experimental data shows $P_g$ dropping slightly below 0.5 at intermediate sweep time, inconsistent with the interpretation that the system population at intermediate sweep time saturates toward the Boltzmann distribution near the minimum gap. This is likely due to imperfect state

preparation at this particular $\Phi_x$ bias, caused by frequency collision with the readout resonator (see Supplementary Note 8). There is significant disagreement between the data and the AME simulation at smaller $\Phi_x$. Specifically, with regard to the $T_{LZ}$ dependence, while the AME predicts $P_g$ to nearly settle at 0.5, the experimental data shows $P_g$ to continue increasing with increasing sweep time. In fact, the experimental final ground state probability $P_g$'s for different $\Phi_x$ crosses at long times and the highest $P_g$ is obtained at the lowest $\Phi_x$. Furthermore, when examining the $\tau$ dependence, it can be seen that for $\tau \gg 1$, where relaxation after the gap is expected to dominate, the simulated $P_g$ curves for different $\Phi_x$ collapse onto the same $\tau$ dependence. This is in contrast with the experimental data, where the $P_g$ curves shift left towards the coherent limit as $\Phi_x$ is reduced. These signatures indicate that the weak coupling limit breaks down as the minimum gap $\Delta$ is reduced.

To understand the data in the strong-coupling limit, we use the polaron-transformed master equation (PTRE)[35,44]. PTRE incorporates strong system-environment coupling by transforming into the dressed polaron frame and treats the tunneling parameter $\Delta$ perturbatively. The noise PSD in the polaron frame is separated into low- and high-frequency components. Particularly, the low-frequency part is characterized by two parameters,

$$W^2 = 2I_p^2 \int \frac{d\omega}{2\pi} S_{\Phi_z, 1/f}^{+}(\omega), \text{ and} \tag{7}$$

$$\epsilon_p = 2I_p^2 \int \frac{d\omega}{2\pi} \frac{S_{\Phi_z, 1/f}^{-}(\omega)}{\hbar\omega}, \tag{8}$$

which are known as the MRT width and reorganization energy respectively[42]. The functions $S_{\Phi_z}^{-(+)}$ are the (anti-) symmetrized low-frequency $\Phi_z$ noise. The PTRE is expected to hold when the environment-induced dephasing is much larger than the qubit's minimum energy gap, or $W \gg \Delta$. By integrating the $1/f$ noise obtained from dephasing time measurements, we obtain $W/h \approx 48$–59 MHz. Assuming the environment is in thermal equilibrium, $\epsilon_p$ is related to $W$ via the fluctuation-dissipation theorem, $\epsilon_p = W^2/2k_BT$. The result of PTRE simulations is shown in Fig. 3e, f. It can be seen that for the nominal noise parameters found above, the final ground-state probability $P_g$ first increases with increasing sweep time, closely

following the coherent LZ probabilities until around $P_g = 0.5$ where it flattens. Through numerical experiments, it is also found that increasing $W$ while keeping $T$ unchanged, or equivalently increasing $\epsilon_p/W$, brings the PTRE results closer to the coherent LZ probabilities. In fact, it has been demonstrated previously that for a flux qubit strongly coupled to low-frequency noise, the LZ transition probability recovers the coherent LZ probability when $\epsilon_p \gg W$[45].

In the experimental data, the transition probabilities approach the coherent limit as $\Phi_x$ and equivalently $\Delta$ is reduced, indicating a strong coupling between the qubit and the environment. At the smallest $\Phi_x$, the measured $P_g$ does not flatten near 0.5, contrasting the PTRE prediction with nominal noise values, but is closer to the PTRE prediction with larger MRT width $W$. This suggests that the noise seen by the qubit is larger than the integrated 1/f noise. This is not entirely surprising, given that previous MRT measurements on superconducting flux qubits also revealed larger $W$ than the integrated power of 1/f flux noise[40,54] (see Supplementary Note 7).

## Spin bath model for the crossover regime

To further understand the result in the crossover regime, we propose a hybrid noise model, that incorporates both the weakly-coupled Markovian noise and the strong low-frequency noise through an explicit spin bath. The motivation for a spin bath is two-fold. First, spin impurities, characterized by logarithmically distributed relaxation times, are one of the main candidates for flux noise on superconducting circuits[41,55–58]. Second, polarization of the bath spins naturally lead to the concept of reorganization energy[58], which is useful in connecting to the strong coupling PTRE model. To keep the model numerically feasible, we consider a spin bath made of $N_s = 3$ spins coupled to the qubit via the interaction term

$$H_{qS} = \sum_i^{N_s} J_i \sigma_z \tau_{z,i},\qquad(9)$$

where $\tau_{z,i}$ is the $i$'th bath spin's Pauli Z operator and $J_i$ is its coupling strength to the qubit. The bath spins do not have internal dynamics, but each of them is transversely coupled to its own environment, at a temperature that we assume to be the same as the qubit's environment, $T$. Each of these secondary environments leads to thermal transitions between the spin states, with depolarization rate $\gamma_i$. For appropriately chosen distributions of $\gamma_i$ and $J_i$, the noise PSD due to the spins effectively represents low-frequency noise with a specified noise amplitude in the chosen frequency range[59] (see Supplementary Note 9).

The spin bath model is simulated using the AME, with the high-frequency noise coupled to the qubit defined in the same way as the single-qubit AME. As a proof-of-concept demonstration, we choose a model with three spins, with uniform ferromagnetic coupling strength $J$, and $\gamma_i$ distributed in the range of 1–10 MHz to match the 1/$f^\alpha$ noise power in this range. The range of $\gamma_i$ corresponds to an intermediate frequency range out of the full frequency span over which the 1/$f^\alpha$ noise PSD is expected to hold, from sub-Hz to about 1 GHz (see Supplementary Note 9). This choice is based on the understanding that noise processes slower than the adiabatic timescale do not affect the transition probability[13], and the dominant effect of fast noise is thermal transitions, which is accounted for by the environment directly coupled to the qubit, as is the case in the single-qubit AME.

As shown in Fig. 4, it is found that if the target 1/f noise amplitude is about 8 times larger than the noise amplitude deduced from decoherence measurements (corresponding to $J_i/h = J/h = -0.09$ GHz), the simulated ground state probabilities versus sweep time at different $\Phi_x$ qualitatively matches the experimental results. Specifically, for large $\Phi_x$ or $\Delta$, the spin bath simulation results display the non-monotonic dependence of ground state probability $P_g$ versus the dimensionless sweep time $\tau$, and is almost indistinguishable from the single qubit AME result. However, at smaller $\Phi_x$ or $\Delta$, the spin bath simulation result displays a monotonic increase of $P_g$ with increasing sweep time $\tau$,

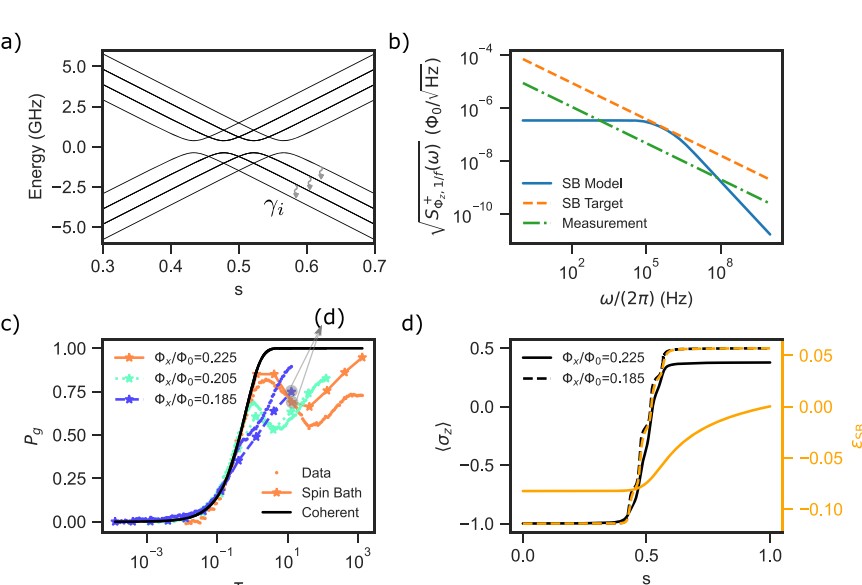

**Fig. 4 | Simulation results of the spin bath model. a** The energy spectra of the qubit going through an LZ transition, with 3 spins ferromagnetically coupled to the qubit, versus the normalized time $s \in [0, 1]$, for $\Phi_x/\Phi_0 = 0.225$. There are 4 distinct levels visible in the manifold of the qubit being in the left state, corresponding to 0, 1, 2, and 3 spins aligned with the qubit (with degeneracies of 1, 3, 3, and 1 respectively). Each of them anti-cross with the corresponding state in the manifold of the qubit being in the right state. Energy levels of states with different numbers of aligned spins cross each other, as there is no matrix element coupling them. The curly arrows indicate the paths for spin relaxation, allowing the spins to align with the qubit after tunneling. **b** The symmetrized 1/$f^\alpha$ flux noise PSD as measured by the dephasing measurement (green dash-dotted line), the equivalent flux noise PSD generated by the spins (blue solid line), calculated by summing the individual Lorentzian contribution of each spin (see Supplementary Note 9), as well as the targeted noise PSD (orange dash line) that the spin bath is set to match, which has 8 times larger amplitude than the measured. **c** The simulated final ground state probabilities versus dimensionless sweep time $\tau$, using the same spin bath parameters as in (**a**, **b**). **d** The evolution of the qubit polarization (black line) and effective bias by the spin bath (orange line), versus the normalized time $s$, for $\Phi_x = 0.185$ (dashed line) and $\Phi_x = 0.225$ (solid line) respectively. Both plots correspond to $\tau \approx 10$ (see text for discussion).

which is similar to the PTRE. The spin bath simulation results also differ from both AME and PTRE in that the $P_g$ curves for different $\Phi_x$ at large $\tau$ do not collapse.

Further intuition about the spin bath model can be obtained by observing the evolution of the polarization of the qubit and the spin bath during the LZ sweep. The collective effect of the spins can be captured by the parameter

$$\epsilon_{SB} = -\sum_i J_i \langle \tau_{z,i} \rangle, \tag{10}$$

which is the effective longitudinal bias applied by the spins on the qubit, analogous to the reorganization energy $\epsilon_p$ in the PTRE model (see Supplementary Note 9). The instantaneous change of $\epsilon_{SB}$ during the sweep is to be compared with the change in qubit polarization $\langle\sigma_z\rangle$, as shown in Fig. 4d for two different values of $X$-flux biases in the adiabatic limit. We can further denote the temporal width of the change of the qubit polarization as the tunneling time $T_t$, which in the adiabatic limit is approximately $\Delta/v$[60]. The tunneling time for the parameter $\Phi_x = 0.225(0.185)$, $T_{LZ} = 10^3(10^5)$ ns is $T_t = 108(995)$ ns. It can be seen that for $\Phi_x = 0.225$, the effective bias $\epsilon_{SB}$ changes slowly after the qubit has tunneled. Therefore the spin bath presents negligible influence on the qubit dynamics. For $\Phi_x = 0.185$, the change in $\epsilon_{SB}$ almost overlaps with the changes in qubit polarization $\langle\sigma_z\rangle$. This is because the spin depolarization rates become shorter as compared to the tunneling time of the qubit as $\Phi_x$ or $\Delta$ reduces. In other words, for small $\Delta$, the spins quickly reorganize themselves to align with the new qubit polarization as the qubit tunnels. This fast change in $\epsilon_{SB}$, together with its relatively large magnitude as compared to $\Delta$, effectively shifts the qubit away from the anti-crossing as soon as the qubit has tunneled to the opposite polarization. Away from the anti-crossing, the qubit rarely experiences thermal excitation, hence the non-monotonic $P_g$ dependence disappears and $P_g$ only increases with sweep time as it has more time to complete the tunneling. If however the noise is not strong and does not induce a large enough $\epsilon_{SB}$ to shift the qubit away from the anti-crossing, thermal excitation would still occur, and $P_g$ would barely increase above 0.5, similar to the single-qubit AME when $\Delta$ is small (see Supplementary Note 9).

## Discussion

The dissipative LZ transition studied here can be considered a toy model for dynamics in quantum annealing, where small gap anti-crossings between the lowest two energy eigenstates are expected to play an important role[18,32]. Our results thus contribute to the understanding of the role of open-system effects in a quantum annealer[28,32,33,61–63]. In the weak coupling limit, thermal relaxation could help the system to reach the ground state. However, for hard problems, the minimum gap is expected to close, which freezes thermal relaxation after the system has passed the minimum gap[32,52,64]. On the other hand, the experiment and the PTRE model both show that higher ground state probabilities can be achieved in the intermediate to strong coupling limit, as compared to the weak coupling limit, when $\epsilon_p/W$ is large enough. However, this is likely specific to the LZ tunneling problem, where the final (right) well becomes much lower than the initial (left) well, allowing tunneling to occur despite the large environment bias preferring the initial well. In the context of hard annealing problems with many quasi-degenerate energy levels, tunneling dynamics of the system could be suppressed due to the environment biasing the energy levels in the strong coupling limit.

The spin bath simulation suggests that only noise that is fast enough as compared to the qubit dynamics will contribute to the strong coupling effect. It is thus expected that in fast annealing, slow noise will not affect the annealing performance, as there is not enough time for a state of definite polarization to be formed. This is in line with the recent demonstration of coherence in fast annealing of flux-qubit-based quantum annealers, on the order of 10 ns, where the

environment is too slow to affect the distribution of final states[63,65]. This immunity to slow noise also points to an important advantage of analog quantum processors over gate-based quantum processors, where in the latter slow noise always causes dephasing and needs to be actively corrected for.

An interesting future direction would be environment engineering techniques to shift the noise at intermediate frequencies to low frequencies, which has been recently demonstrated by applying a magnetic field in-plane to the superconducting circuit[41]. Further development of such techniques likely requires an improved understanding of the physical origins of flux noise. It should be noted that in the simulation, the noise power required to obtain agreement with the experiment is larger than estimated from dephasing time measurements. This could be due to the sensitivity of the measurement to different noise frequencies, and the non-Gaussian nature of the low-frequency noise. Future experiments could use different control protocols, such as repeated LZ crossing, and locally adiabatic control protocols, which likely provide sensitivity to noise at different frequencies. These experiments, in combination with noise spectroscopy techniques[39], could be used to further settle the actual noise power seen by the qubit, and elucidate the physical origins of low-frequency flux noise.

In summary, we experimentally characterized the LZ transition probability in a superconducting flux qubit with a wide range of sweep velocities $v$ and minimum gap sizes $\Delta$, and we showed a crossover from weak to strong coupling to flux noise. We found that for large gap $\Delta$, the competition between adiabaticity and environment-induced thermalization leads to non-monotonic dependence of the final ground state probability $P_g$ on sweep time, which can be reproduced by a weak coupling model, the AME. However, as $\Delta$ becomes smaller, the non-monotonicity gradually disappears and $P_g$ becomes closer to the coherent LZ transition rate, which is consistent with a strong-coupling model, the PTRE. We also explored a spin bath model that qualitatively reproduces the full range of experimental data. The spin bath model explicates that the crossover depends on the relative timescale between the tunneling time of the qubit, and the time taken for the environment to reorganize itself after the qubit has tunneled. Our work brings insights into the role of low-frequency noises in quantum tunneling, which is relevant to quantum annealing and more broadly tunneling phenomena in quantum chemistry and biology. Future extensions of this work include direct measurement of the environment reorganization energy and timescale in different control protocols and the investigation of the crossover region in multi-qubit settings.

## Methods

### Device fabrication

The device is fabricated at MIT Lincoln Laboratory using a multi-tier fabrication process, which consists of a high-coherence qubit tier, an interposer, and a multi-layer control tier[66,67]. Our device only uses the qubit and interposer tiers. The qubit tier contains high-quality Al forming the qubit loop and capacitance, as well as Al/AlO$_x$/Al junctions. The interposer tier contains the tunable resonator and the control lines. They are bump-bonded together in a flip-chip configuration using indium bumps.

### Qubit control and readout

The qubit $x$, $z$ flux biases are each controlled using on-chip flux bias lines, with current supplied by a DC voltage source and a fast AWG. The DC source has a stronger coupling to the qubit loops to achieve flux biasing of more than a flux quantum. The AWG has weaker coupling to the flux biases, corresponding to a bias range of about 100 m$\Phi_0$.

Before the LZ sweep, the qubit needs to be prepared in its ground state. Passive cooling is not possible when the tunneling barrier between persistent current states is large. We use a cooling protocol

similar to that demonstrated in ref. 68. Using the large tunneling amplitude between the ground state of the higher well and the excited state of the opposite well, residual excited populations can be adiabatically transferred to the lower well and the system quickly relaxes to the ground state. Additional details of the energy spectrum and this cooling method are discussed in Supplementary Note 6.

The qubit state is read by measuring the state-dependent transmission through the tunable resonator. When doing spectroscopy and coherence measurements, the resonator is biased at a flux-insensitive position (0 flux in the SQUID), and readout is in the qubit energy eigenbasis, due to dispersive interaction between the qubit and resonator. During the LZ measurement, the resonator is biased at a flux-sensitive position ($-0.15\,\Phi_0$ flux in the SQUID), which allows measuring the qubit's persistent current states. Since the readout point of the LZ experiment is far from the qubit's symmetry point, the persistent current basis and energy eigenbasis of the qubit nearly coincide, with more than 99% overlap, and we do not distinguish the two bases at the readout point.

### Noise characterization

The qubit is capacitively coupled to a microwave line. This allows spectroscopy, qubit relaxation (T1), and qubit dephasing (T2) measurement when $\Delta/h \gtrsim 1\,\text{GHz}$. The flux-dependent qubit transition frequencies are used to fit a complete circuit model of the device, which is used to compute $I_p$ and $\Delta$ at different $\Phi_x$[69]. The circuit model is also used to compute the noise sensitivity of the qubit at different flux biases. Combining the noise sensitivity with the T1, T2 measurement results allows us to determine the noise PSD, which is used in the master equation simulation for LZ transition. More details of the noise characterization are discussed in a separate publication[50]. The noise parameters used are given in Supplementary Note 7.

### Master equation simulation

The master equation simulations were performed using HOQST, a Julia package for open-system dynamics with time-dependent Hamiltonians[44]. The simulation takes the qubit Hamiltonian in Eq. (2), with $I_p$, $\Delta$ given by the circuit model. The qubit-bath interaction considered is

$$H_{qb} = -I_p \sigma_z \otimes Q_{\Phi_z}, \tag{11}$$

where $\sigma_z$ acts on the qubit and $Q$ acts on the environment degrees of freedom which causes the qubit $z$ loop flux noise.

The AME is a time-dependent version of the frequency-form Lindblad equation. Although a rigorous upper bound on the error of the AME is only small in the adiabatic limit[70], it is likely that the error is still small in the non-adiabatic limit, as shown in previous work where the master equation results are compared to numerically exact path integral based simulation results[20,53]. Specifically, the form used in this work is the one-sided AME that first appeared in[52], given by

$$\dot{\rho}_q(t) = -\frac{i}{\hbar}[H_q(t), \rho_q(t)] \\ + \frac{1}{\hbar^2}\sum_\omega \Gamma_{\Phi_z}(\omega)\left[L_\omega(t)\rho_q, I_p\sigma_z\right] + h.c.,$$

where $\rho_q$ is the reduced density matrix of the qubit,

$$\Gamma_{\Phi_z}(\omega) = \frac{1}{2}S_{\Phi_z}(\omega) + i\gamma_{\Phi_z}(\omega) \tag{12}$$

$$\gamma_{\Phi_z}(\omega) = \frac{1}{2\pi}\int_{-\infty}^{\infty} S_{\Phi_z}(\omega')\mathcal{P}\left(\frac{1}{\omega - \omega'}\right)d\omega', \tag{13}$$

with $\mathcal{P}$ denoting the Cauchy principal value. The operators $L_\omega$ are given by

$$L_\omega(t) = \sum_{E_\beta - E_\alpha = \omega} \langle\alpha(t)|I_p\sigma_z|\beta(t)\rangle\,\alpha(t)\langle\beta(t)|, \tag{14}$$

where $E_{\alpha(\beta)}$ and $|\alpha(\beta)\rangle$ are the system's instantaneous energy eigenvalues and eigenstates, and $\alpha, \beta \in \{g, e\}$.

The PTRE is a model that accounts for strong system-environment coupling and has been found to explain experimental data in quantum annealers coupled strongly to low-frequency noise[35,45]. The form of PTRE we use here is the Lindblad form, given by

$$\dot{\tilde{\rho}}_q(t) = -\frac{i}{\hbar}[\tilde{H}_q(t), \tilde{\rho}_q(t)] \\ + \frac{1}{\hbar^2}\sum_{\omega,\lambda}\tilde{S}(\omega)\left[\tilde{L}_{\omega,\lambda}\tilde{\rho}_q\tilde{L}_{\omega,\lambda}^\dagger - \frac{1}{2}\left\{\tilde{L}_{\omega,\lambda}^\dagger\tilde{L}_{\omega,\lambda}, \tilde{\rho}_q\right\}\right],$$

where tilde is used to denote operators in the polaron frame. Specifically,

$$\tilde{H}_q = -I_p(\Phi_z - \Phi_z^{\text{sym}})\sigma_z, \tag{15}$$

$$\tilde{L}_{\omega,\lambda\in\{+,-\}} = \frac{\Delta}{2}\sum_{E_i - E_j = \hbar\omega}\langle i|\sigma_\lambda\,|\,j\rangle|i\rangle\langle j|, \tag{16}$$

where $\sigma_{+(-)}$ is the qubit Pauli raising and lowering operators and $i, j \in \{0, 1\}$ are the qubit persistent current state index. The low- and high-frequency parts of the noise give a convolutional form for the PSD in the polaron frame

$$\tilde{S}(\omega) = \hbar^2\int\frac{d\omega'}{2\pi}G_L(\omega - \omega')G_H(\omega'), \tag{17}$$

where $G_L$ and $G_H$ are contributions from the low- and high-frequency noise respectively, given by

$$G_L(\omega) = \sqrt{\frac{\pi}{2\hbar^2 W^2}}\exp\left[-\frac{(\hbar\omega - 4\epsilon_p)^2}{8\hbar^2 W^2}\right] \tag{18}$$

$$G_H(\omega) = \frac{4S_{\Phi_z,\text{ohmic}}(\omega)I_p^2}{\hbar^2\omega^2 + 4S_{\Phi_z,\text{ohmic}}(0)I_p^2}. \tag{19}$$

### Spin bath simulation

In the spin bath model, the Hamiltonian of the system qubit, the spin bath, together with their respective environment is given as

$$H = H_q + H_S + H_{qS} + H_{qb} + H_{SB} + H_b + H_B, \tag{20}$$

with

$$H_{qS} = \sum_i^{N_s} J_i\sigma_z\tau_{z,i}, \tag{21}$$

$$H_S = 0, \tag{22}$$

$$H_{SB} = \sum_i^{N_s}\tau_{x,i}Q_i', \text{ and} \tag{23}$$

$$S_{Q_i'}(\omega) = \hbar^2 \lambda_i \frac{1}{1 + \exp(-\beta\hbar\omega)} \exp\left(-\frac{\omega}{\omega_c}\right). \quad (24)$$

Here $H_b$ is the qubit environment and $H_B$ is the secondary environment coupled to the spins. The operator $Q_i'$ is an operator of the $i'$ th secondary environment that is coupled to spin $i$. The noise PSD of the $Q_i'$ operator is nearly white noise, with strength $\lambda_i$, inverse temperature $\beta$, and cutoff frequency $\omega_c$ assumed to be the same as the qubit high-frequency environment. This can be thought of as a quantum extension of simulating classical $1/f$ noise with two-level fluctuators[36], The crucial difference here that distinguishes these spins from classical two-level fluctuators is that the relaxation and excitation rates obey detailed balance. Indeed, if the spins are sufficiently weakly coupled to the qubit and the energy difference between the spin up and down states are zero (or if the spin environment has infinite temperature), the spin has the same relaxation and excitation rate, $\lambda/2$, acting exactly like a classical two-level fluctuator.

The spin bath parameters $J_i$ and $\lambda_i$ are chosen based on the measured $1/f$ flux noise strength. Specifically, the noise strength $\lambda_i$ determines the spin depolarization rate and is chosen based on the range of noise frequency of interest. To determine the coupling strength $J_i$, we consider the effective noise PSD contributed by each spin[59], given by

$$S_i(\omega) = (1 - \langle\tau_i\rangle)\frac{2\gamma_i J_i^2}{\omega^2 + \gamma_i^2}, \quad (25)$$

where $\langle\tau_i\rangle$ is the expectation value of spin $i$'s longitudinal polarization. The coupling strength $J_i = J$ is determined by summing the PSDs of individual spins and comparing with the target noise amplitude. We note that the spin's effective PSD is strictly only valid when it is weakly coupled to the qubit. However, it serves as a useful guide in choosing the spin bath parameters. Further simulation of the spin bath model and its comparison with the MRT parameters are given in the Supplementary Note 9.

## Data availability

The data generated in this study have been deposited in the Figshare database, which can be accessed via the link https://doi.org/10.6084/m9.figshare.26531362[71].

## Code availability

Analysis and simulation codes that support the findings of this study are available from the corresponding authors upon request.

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

## Acknowledgements

We thank the members of the Quantum Enhanced Optimization (QEO)/ Quantum Annealing Feasibility Study (QAFS) collaboration for various contributions that impacted this research. In particular, we thank D. Ferguson for fruitful discussions of experiments, A. J. Kerman for guidance on circuit simulations and design, and R. Yang for related work on circuit modeling and useful discussions. We also thank fruitful discussions with W. Strunz, V. Link, F. S. Kahlert, and D. Segal on the open quantum system models. We gratefully acknowledge the MIT Lincoln Laboratory design, fabrication, packaging, and testing personnel for valuable technical assistance. The theoretical modeling of the work benefited from the high-performance computing cluster provided by SHARCNET (sharcnet.ca) and the Digital Research Alliance of Canada (alliancecan.ca). The research is based upon work supported by the Office of the Director of National Intelligence (ODNI), Intelligence Advanced Research Projects Activity (IARPA), and the Defense Advanced Research Projects Agency (DARPA), via the U.S. Army Research Office contract W911NF-17-C-0050. The views and conclusions contained herein are those of the authors and should not be interpreted as necessarily representing the official policies or endorsements, either expressed or implied, of the ODNI, IARPA, DARPA, or the U.S. Government. The U.S. Government is authorized to reproduce and distribute reprints for Governmental purposes notwithstanding any copyright annotation thereon.

## Author contributions

R.T. performed the experiments. R.T. and X.D. performed the data analysis. X.D. performed the numerical simulations. H.C. provided guidance on the numerical simulations and theoretical models. D.M., M.A.Y., A.J. Martinez, and Y.T. designed the device. S.N. provided feedback on the device design. E.M. explored alternative explanations of the experimental data within the AME model. J.G., J.A.G., and X.D. performed earlier versions of the experiments on a different device and setup. D.M.T., J.A.G., S.D., and J.B. contributed to the development of the experiment infrastructure. R.D., A.J.Melville, B.M.N., and J.L.Y. developed the fabrication process and fabricated the device. C.H., K.S., and S.J.W. contributed to the fridge and electronics operation. W.O. supervised the QEO/QAFS effort from Lincoln lab, K.M.Z. led the coordination of the QEO/QAFS experimental effort, D.L. led the QEO/QAFS program and A.L. proposed and supervised this work. All authors were involved in the discussion of experiments and data analysis. X.D., R.T., and A.L. wrote the paper with feedback from all authors.

## Competing interests

The authors declare no competing interests.
