## [Transparent Peer Review file · Nature Communications]

Dissipative Landau-Zener tunneling in the crossover regime from weak to strong environment coupling

Corresponding Author: Dr Xi Dai

Version 0:

Reviewer comments:

Reviewer #1

(Remarks to the Author)

Dear editor,

this manuscript reports an experimental study on Landau Zener transitions in the presence of a dissipative environment studied as a function of minimum gap and sweep time. A non monotonic behaviour as a function of sweep time connected to distinct effect of low and high frequency noise on the dynamics is observed. The authors in the new version of the manuscript present a toy model reproducing qualitatively the main features of the data. The model, despite being schematic, has some realistic elements considering the fact that the dominant source of dephasing in flux qubits are two level impurities.

I read carefully the reply as well as the manuscript and in particular the new sections and in the supplementary material addressing the issues that I raised in my report. I find the reply satisfactory and I therefore recommend the manuscript for publication in Nature Communications.

Reviewer #2

(Remarks to the Author)

The paper presents a study of the dissipative Landau-Zener effect using a flux qubit, where the gap can be modified by using a magnetic flux. Although there has been considerable theoretical effort by the authors to model the noise in this system (and with reasonably good agreement with the data), I do not quite see the importance of what has been achieved here. I find it unsurprising that there is a crossover as one changes the gap - what does it prove or what does it allow us to do? If somehow there would be a control or engineering of the environment maybe that would make things more interesting, but the reality is that this seems a low-quality sample (to my surprise, notwithstanding all these sophisticated spin and ohmic models for decoherence, not a single T1 or T2 number is given in the paper or supplement, but the Ramsey figure from the supplement gives some idea) where this study was all that can be done. Perhaps the paper could be transferred to Communication Physics.

Response to reviewers

Our response is added in blue fonts.

Reviewer #1 (Remarks to the Author):

Dear editor,

this manuscript reports an experimental study on Landau Zener transitions in the presence of a dissipative environment studied as a function of minimum gap and sweep time. A non monotonic behaviour as a function of sweep time connected to distinct effect of low and high frequency noise on the dynamics is observed. The authors in the new version of the manuscript present a toy model reproducing qualitatively the main features of the data. The model, despite being schematic, has some realistic elements considering the fact that the dominant source of dephasing in flux qubits are two level impurities.

I read carefully the reply as well as the manuscript and in particular the new sections and in the supplementary material addressing the issues that I raised in my report. I find the reply satisfactory and I therefore recommend the manuscript for publication in Nature Communications.

We appreciate the reviewer's thorough examination of the revised manuscript and response, and we are grateful for the recommendation for publication.

Reviewer #2 (Remarks to the Author):

The paper presents a study of the dissipative Landau-Zener effect using a flux qubit, where the gap can be modified by using a magnetic flux. Although there has been considerable theoretical effort by the authors to model the noise in this system (and with reasonably good agreement with the data), I do not quite see the importance of what has been achieved here. I find it unsurprising that there is a crossover as one changes the gap - what does it prove or what does it allow us to do? If somehow there would be a control or engineering of the environment maybe that would make things more interesting, but the reality is that this seems a low-quality sample (to my surprise, notwithstanding all these sophisticated spin and ohmic models for decoherence, not a single T_1 or T_2 number is given in the paper or supplement, but the Ramsey figure from the supplement gives some idea) where this study was all that can be done. Perhaps the paper could be transferred to Communication Physics.

We thank the referee for reviewing our manuscript and appreciating the addition of the new theory model, which is in agreement with the data.

We agree with the reviewer on the point that a qualitative change in the Landau Zener probability versus minimum gap can be expected based on general

considerations and in line with the previous investigations of Landau Zener dynamics in the limits of very weak and very strong coupling. Our work was specifically motivated by the presumed existence of such a crossover and, very importantly, the perspective of understanding the mechanisms and the model parameters that determine this crossover. Our analysis and theoretical modelling demonstrate the importance of not just the strength of the noise, but also the correlation time of the noise. We agree with the reviewer that the exploration of this crossover in combination with environment engineering is interesting and we hope that the insights from our work could guide future efforts in this direction.

We respectfully disagree with the referee that this is low-quality sample. From coherence measurements, we infer that the coherence is intrinsic flux noise limited, with flux-noise amplitude similar to the noise magnitude in previously reported devices. The large susceptibility to flux-noise is a design decision, which allows strong coupling between qubits (on the order of 1GHz, see e.g. Phys. Rev. Appl. 8, 014004 (2017)), which is essential for quantum annealing applications based on such qubits.

We thank the referee for pointing out the lack of T1 and T2 numbers in the paper. For our device, coherence time results vary greatly depending on the flux biasing point, and are only valid if the coupling to the environment is sufficiently weak. In the flux bias range where the Landau-Zener measurements are performed, coherence measurements are difficult due to the relatively strongly coupled environment. However, coherence measurements of this device in the weak-coupling regime are discussed in detail in a separate manuscript (arXiv:2307.13961). These results were previously only referenced in the Supplementary Information. We have now added this reference into the main text. In addition, we added a figure in the Supplementary Information to present typical coherence time numbers in the weak-coupling limit.

Summary

Landau-Zener transitions near an avoided level crossing is a broadly relevant paradigm in science. An experiment using a flux qubit sheds new light on the role of the environment, through the observation of a crossover between weak and strong coupling.